

# Catchments catch all in South African coastal lowlands: topography and palaeoclimate restricted gene flow in *Nymania capensis* (Meliaceae)— a multilocus phylogeographic and distribution modelling approach

Alastair J. Potts

Department of Botany, Nelson Mandela Metropolitan University, Port Elizabeth, South Africa
Department of Biological Sciences, University of Cape Town, Cape Town, South Africa

## ABSTRACT

**Background**. This study investigates orbitally-forced range dynamics at a regional scale by exploring the evolutionary history of *Nymania capensis* (Meliaceae) across the deeply incised landscapes of the subescarpment coastal lowlands of South Africa; a region that is home to three biodiversity hotspots (Succulent Karoo, Fynbos, and Maputaland-Pondoland-Albany hotspots).

**Methods**. A range of methods are used including: multilocus phylogeography (chloroplast and high- and low-copy nuclear DNA), molecular dating and species distribution modelling (SDM).

**Results**. The results support an 'evolutionarily distinct catchment' hypothesis where: (1) different catchments contain genetically distinct lineages, (2) limited genetic structuring was detected within basins whilst high structuring was detected between basins, and (3) within primary catchment populations display a high degree of genealogical lineage sorting. In addition, the results support a glacial refugia hypothesis as: (a) the timing of chloroplast lineage diversification is restricted to the Pleistocene in a landscape that has been relatively unchanged since the late Pliocene, and (b) the projected LGM distribution of suitable climate for *N. capensis* suggest fragmentation into refugia that correspond to the current phylogeographic populations.

**Discussion**. This study highlights the interaction of topography and subtle Pleistocene climate variations as drivers limiting both seed and pollen flow along these lowlands. This lends support to the region's large-scale conservation planning efforts, which used catchments as foundational units for conservation as these are likely to be evolutionarily significant units.

Corresponding author
Alastair J. Potts, potts.a@gmail.com

# INTRODUCTION

Climate changes driven by variations in the Earth's orbit, i.e., Milankovitch oscillations, have influenced the distribution of species and clades across the globe. *Jansson & Dynesius (2002)* termed this "orbitally-forced range dynamics" (ORD) and highlighted that this process drives extinction, splitting, and merging of gene pools and clades. The magnitude of ORD varies geographically, primarily latitudinally with greater ORD in the poles and a decline towards the equator (*Dynesius & Jansson, 2000*). Gene pools should persist without going extinct or merging in areas with low ORD, leading to long term splitting and diverging of clades; *Jansson & Dynesius (2002)* call this $\beta$-clade formation where evolutionary change persists for more than 100 kyr. Here I explore the role of ORD on a regional scale, specifically in terms of the interaction between glacial–inter-glacial climate changes and topography (i.e., watershed configuration) on a tree species, *Nymania capensis* (Meliaceae), across a section of the subescarpment coastal plains in South Africa.

*Nymania capensis* is associated with the subtropical thicket vegetation in South Africa, locally termed the Albany Subtropical Thicket. This region forms the western part of the Maputaland-Pondoland-Albany biodiversity hotspot (*Steenkamp et al., 2004*). Thicket vegetation is characterised as dense, woody, semi-succulent and thorny, with an average height of 2–3 m, and is relatively impenetrable in a pristine condition (*Vlok, Euston-Brown & Cowling, 2003*). It is largely restricted to the subescarpment coastal plains along the south coast of South Africa where it spans a number of deeply incised primary catchments. Although thicket has a continuous or near-continuous distribution across catchments, each catchment has been identified as a discrete biogeographic unit (*Vlok, Euston-Brown & Cowling, 2003*) suggesting that the watersheds have been dispersal barriers to thicket plant species. This pattern, termed here as the "evolutionarily discrete catchment" (EDC) hypothesis, has formed the foundation of large-scale conservation planning that aims to ensure the persistence of evolutionary processes for thicket biota (*Rouget et al., 2006*). This hypothesis can be seen as a specific case of low ORD where the topographic heterogeneity of the deeply incised subescarpment lowlands coupled with climate stability has enabled $\beta$-clade formation, i.e., clades that have survived and remained genetically separate for longer than 100 kyr. Under the EDC hypothesis, a number of predictions can be generated if catchments and their associated watersheds are important landscape features responsible for structuring genetic diversity (sensu *Price, Barker & Villet, 2010*; *Potts et al., 2013b*):

(1) Different catchments should contain genetically distinct lineages,

(2) There should be limited genetic structuring within catchments and a high degree of genetic structuring among basins, and

(3) If watersheds have been long term barriers to gene flow, then isolated catchment populations should contain evidence of genealogical sorting.

Southern Africa did not experience glaciation during the Pleistocene climate cycles (*Partridge, 1997*). Nonetheless, these watershed barriers are suggested to have been greatly strengthened during the Pleistocene as thicket is predicted to have severely contracted into, and fragmented across, catchments (*Cowling, Proches & Vlok, 2005*; *Potts et al., 2013b*): a glacial multiple refugia (GMR) hypothesis (a low ORD scenario). If thicket species were

adversely affected by glacial climates through the Pleistocene, in interaction with the topography, then the following is predicted:

(4) The timing of population subdivision between the catchments should coincide with the onset of glacial–interglacial cycles as these catchments are largely fixed and stable landscape features through this period (*Cowling, Proches & Partridge, 2009*), but also likely form *β*-clades that have persisted for longer than 100 kyr (*Jansson & Dynesius, 2002*), and

(5) The species' distribution should experience contraction and fragmentation that is consistent with phylogeographic patterns.

There is growing support that these EDC-GMR hypotheses apply to thicket vegetation (*Duker et al., 2015a*; *Potts et al., 2013b*). Plastid lineages from thicket taxa *Nymania capensis* and *Pappea capensis* have previously been shown to be isolated to primary catchments (*Potts, Hedderson & Cowling, 2013*). However, being strongly geographically constrained, plastid markers may give a biased representation of a species' genetic coherence (*Premoli et al., 2012*) or evolutionary history, as the smaller effective population size of uniparentally-inherited markers render them more susceptible to stochastic processes (*Edwards & Beerli, 2000*). Therefore, this study provides an in-depth phylogeographic examination of *Nymania capensis* by combining the results from *Potts, Hedderson & Cowling (2013)* with data from high and low copy nuclear regions. These data are used to test the EDC hypothesis by determining whether populations within populations are evolutionarily significant units (sensu *Moritz, 1994*). In addition, molecular dating of the chloroplast dataset and hindcasting of species distribution models are used to explore the GMR hypothesis. The results are contrasted with the predictions generated from these two hypotheses.

## MATERIALS AND METHODS

### Study system

Thicket is restricted to the year-round rainfall zone along the coastal lowlands of South Africa. A winter rainfall zone is found to the west and a summer rainfall zone to the east. The subescarpment coastal lowlands form a series of short but deeply incised catchments separated from an unusually elevated interior plateau by the Great Escarpment. The coastal lowland landscape has been topographically stable and relatively unchanged since the end of the Pliocene (∼2.6 Ma; *Cowling, Proches & Partridge, 2009*) and possibly even longer (*Scharf et al., 2013*).

The distribution of *N. capensis* in the Albany Subtropical Thicket spans three primary catchments along the coastal lowlands, specifically the Gouritz, Gamtoos and Sundays (named after the major rivers within each catchment). However, the Gouritz catchment has additional topographic complexity as two parallel mountain ranges associated with the Cape Fold Belt run across it, creating an intermontane basin known as the Little Karoo. An inselberg, the Rooiberg Mountain, splits this intermontane basin along a west-east orientation (see Fig. 1A). This inselberg has been found to be a barrier to gene flow in another terrestrial plant species endemic to the Little Karoo (*Potts et al., 2013a*), so the secondary catchments west and east of the Rooiberg (Groot and Olifants, respectively) were sampled as extensively as the primary catchments.
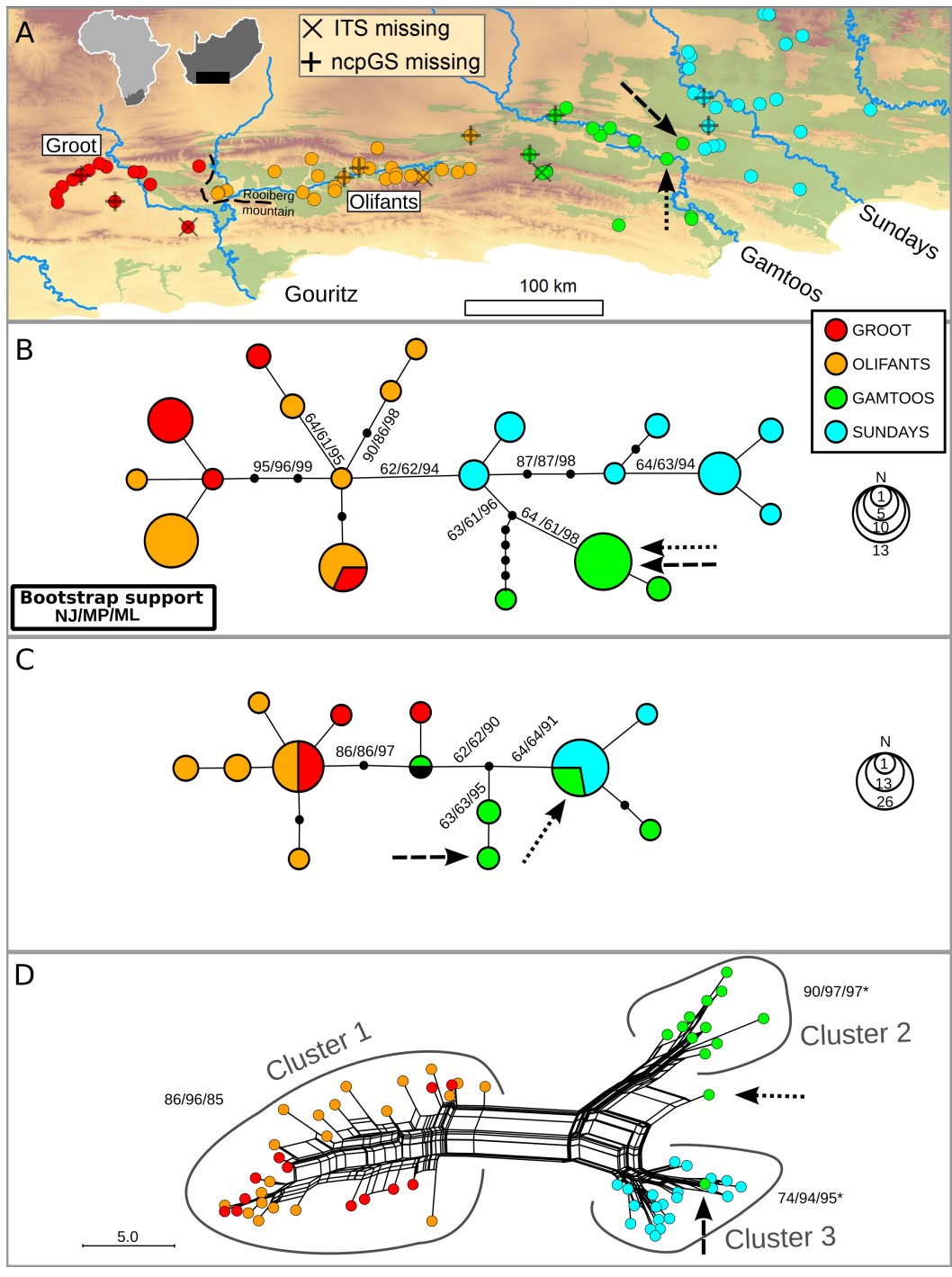

**Figure 1** Distribution of *Nymania capensis* sampling localities within the Albany Subtropical Thicket (light green) (A), and evolutionary relationships among accessions based on three different molecular regions: cpDNA (B), low-copy nDNA (ncpGS) (C), and high-copy nDNA (ITS) (D). Arrows represent samples AJP0537 and AJP0810, respectively, which are discussed in the text. Outgroup samples were pruned from these networks.

## Study species

*Nymania* is a monotypic, woody genus of Meliaceae restricted to southern Africa (see Appendix S1 for images of the growth habit, flowers and inflated seed capsules). Solitary flowers are borne on leaf axils and are generally pollinated by insects (JHJ Vlok & AJ Potts, pers. obs., 2007). The fruit is an inflated and deeply lobed capsule with papery thin membranes, and contains numerous small seeds. The seeds are carried away from the parent plant within the light inflated capsules that are blown along the ground by wind or carried along by water. The inflated capsule is peculiar in this species as it is not described anywhere else in the Meliaceae (*Pennington & Styles, 1975*).

This species occurs across a wide range of environmental conditions and is split into two disjunct distributions, one in the southern semi-arid region (∼300–600 mm annual rainfall) of the Albany Subtropical Thicket and the other in the northern arid region (∼100–300 mm annual rainfall) of South Africa, which is a mosaic of Subtropical Thicket and Nama-Karoo vegetation. Preliminary morphological (AE Van Wyk, pers. comm., 2009) and genetic data (two samples used in this study) suggest that the northern distribution that extends from the Nama-Karoo into Namibia and the southern distribution, which is restricted to the Albany Thicket, are different species. The northern and southern distributions of *N. capensis* are separated by a distance of over 400 km spanning the Great Escarpment mountain range and the interior elevated plateau of South Africa. This species was selected because (i) it is a common component of the Albany Thicket, (ii) for its ease of location and identification in the landscape, and (iii) human activities, such as livestock farming, do not appear to have affected its local distribution in the landscape (AJ Potts, pers. obs., 2007–2009).

## Samples and DNA data

Sampling consisted of one individual from each of 78 localities across the thicket. Sampling permits were obtained from the Eastern Cape Parks and Tourism Agency and CapeNature (0028-AAA008-00015). The sampling strategy was focussed on maximising the number of sites sampled in order to explore the broad scale regional patterns rather than intra-population differences. This scattered sampling strategy is also not affected by local and rapid coalescence events (*Städler et al., 2009*) and thus gives an unbiased view of population structuring and demographic history. Thus, each catchment was represented by 14 to 26 individuals, with nearly all individuals selected from localities that were over 10 km apart. Two samples of *N. capensis* from the northern distribution stored in the Bolus Herbarium were used to generate outgroup accessions (Bolus Herbarium voucher numbers BOL48535 and BOL60966).

Three DNA sequence markers were analysed: chloroplast DNA and nuclear DNA from high and low copy regions. Chloroplast DNA accessions of the trnQ-5′–rps16 and atpI–atpH regions (from *Potts, Hedderson & Cowling, 2013*) are treated as linked regions as they are inherited in tandem. Nuclear DNA sequence data were obtained from two loci, the high-copy ITS region of the 35S rDNA cistron (the 5.8S rDNA and the flanking internal transcribed spacers, ITS-1 and ITS-2 from *Potts, Hedderson & Grimm, 2014*, complemented

with additional accessions) and the low-copy chloroplast-expressed glutamine synthetase gene (ncpGS; new data). Extraction and PCR protocols are reported in Appendix S2.

Details of sequence alignment and, in the case of nuclear DNA, the detection of intra-individual site polymorphisms (2ISPs) are given in *Potts, Hedderson & Grimm (2014)*. In brief, multiple variants of a nuclear region may be present within an individual which give rise to 2ISPs. These can be observed as double (and even triple) peaks in trace files and were coded using the extended IUPAC nucleotide codes (e.g., Y, R); these codes are commonly referred to as 'ambiguity' codes, but it is important to note that 2ISPs are not the result of uncertain coding due to scrappy trace files but rather the clear presence of two or more bases. Numerous 2ISPs were observed in direct-PCR sequences from both nuclear regions, thus indicating the presence of more than one DNA variant present in the genome. This has been observed in many other plant species (*Bailey et al., 2003*), and is unsurprising given the complexity of the nuclear regions, especially ITS.

How accurate is direct-PCR sequencing for detecting 2ISPs? This question has not been investigated in-depth in this study or elsewhere. However, using a small subset of *N. capensis* samples, I found a high degree of congruence between direct-PCR and cloned samples with a low false detection rate; two other studies show similar findings (see Appendix S2 for further details). Both ITS and ncpGS cloned samples demonstrated that these regions contained multiple copies of each region per sample (see Supplementary Data Files). This rules out statistically inferring phased haplotypes from polymorphic sequences as these methods assume a maximum of two copies per individual (e.g., PHASE, *Stephens, Smith & Donnelly, 2001*) and cloning all samples was prohibitively expensive.

All datasets aligned readily and reliable gaps corresponding to insertion or deletion events were included as informative characters; indels that were not associated with homopolymer repeats were coded as binary characters and used in all analyses (multiple base indels were treated as single characters). Where 2ISPs comprised an indel and a base or bases (observed in the ITS dataset), then these were given a unique code for inclusion into subsequent analyses as no IUPAC nucleotide code exists for this combination.

## Phylogenetic and phylogeographic analyses

The phylogenetic relationships between chloroplast, ncpGS haplotypes and all ITS ribotypes were inferred using Statistical Parsimony (SP) and NeighbourNet (NN) networks in TCS version 1.21 (*Clement, Posada & Crandall, 2000*) and SplitsTree version 4.8 (*Huson & Bryant, 2006*), respectively. In order to accommodate the 2ISPs present in the ncpGS and ITS datasets, polymorphism $p$-distances (*Potts, Hedderson & Grimm, 2014*) were used to calculate both SP and NN networks; this measure specifically incorporates 2ISPs when calculating genetic distances (these sites are usually removed or averaged) and was obtained using the PHANGORN library (*Schliep, 2011*) in R3.0.1 (*R Core Team, 2015*). Consensus networks generated from 1,000 bootstraps using Neighbour Joining, Maximum Parsimony or Maximum Likelihood were used to assess edge support in the SP and NN networks; details of software used and settings are given in *Potts, Hedderson & Grimm (2014)* but, in brief, PAUP* version 4b10 (*Swofford, 2002*) and RAxML version 7.2.6 (*Stamatakis, 2006*; *Stamatakis, Hoover & Rougemont, 2008*) were used under the 2ISP-informative approach.

The standard treatment of 2ISPs by phylogenetic software is that the actual nucleotide is uncertain (or 'ambiguous') and this usually leads to an erosion of phylogenetic support. The 2ISP-informative approach avoids this by forcing the phylogenetic algorithms to treat 2ISPs as additional information (*Potts, Hedderson & Grimm, 2014*).

Bayesian software such as MrBayes (*Ronquist & Huelsenbeck, 2003*) or BEAST (*Drummond & Rambaut, 2007*) cannot incorporate 2ISPs as informative characters (*Potts, Hedderson & Grimm, 2014*) and return unresolved topologies for the ITS and ncpGS alignments (results not shown).

The level of genealogical divergence was assessed using the genealogical sorting index (*gsi*, *Cummings et al., 2008*). The *gsi* statistic is a standardised measure of the extent to which predefined groups in a gene tree exhibit exclusive ancestry (fall within a distinct clade); the *gsi* statistic ranges from 0 (a complete lack of genealogical divergence with respect to other groups) to 1 (monophyly: all elements of a predefined group fall within the same, exclusive subtree). Further details are provided in Appendix S2.

Pairwise genetic and geographic distances among sampling locations within and between catchments were used to test patterns of isolation by distance using a Mantel test (*Mantel, 1967*). Genetic distance for cpDNA was calculated using uncorrected *p*-distances, whereas ITS and ncpGS genetic distances were calculated using polymorphism *p*-distances (*Potts, Hedderson & Grimm, 2014*). Standard genetic and geographic distances were calculated using the APE library version 3.0-9 in R (*Paradis, Claude & Strimmer, 2004*). The probability and significance of the correlation coefficient (Spearman's R) was estimated after 10,000 permutations.

## Molecular dating

Given the dominance of 2ISPs in the nuclear datasets, only the cpDNA could be used for molecular dating; a molecular clock approach using such individual consensus sequences is not yet feasible and the alternative 2ISP resampling procedure suggested by *Lischer, Excoffier & Heckel (2013)* generates exceptionally wide and uninformative date ranges due to the high proportion of 2ISPs in the ITS dataset (results not shown). Assigning a timescale to phylogenies typically involves ingroup fossils as age constraints or applying a rate of molecular evolution. No fossil or pollen data specific to *N. capensis* are available, so the hypothesis of Pleistocene lineage evolution was tested using a highly conservative approach by using the 'extremes' of substitution rates found in the literature for non-coding chloroplast DNA, specifically $1.0 \times 10^{-9}$ (*Richardson et al., 2001*) and $31 \times 10^{-9}$ substitutions per site per year (*Fu & Allaby, 2010*). The dating of lineage divergence was carried out using Beast version 1.4.8 (*Drummond & Rambaut, 2007*), which estimates the tree topology and the date (height) of nodes simultaneously using a Bayesian approach. The Beast analysis was performed using all cpDNA samples as this sampling is important for coalescent estimation of lineage divergence. Further details are provided in Appendix S2.

## Distribution modelling

The details of the species distribution modelling approach, including hindcasting the models to Last Glacial Maximum climate conditions are given in *Potts, Hedderson &*

*Cowling (2013)*. In brief, an ensemble species distribution modelling approach (sensu *Araújo & New, 2007*) was used where the uncertainty generated by locality selection, parameter selection of current climate and distribution modelling algorithm could be assessed; this resulted in an ensemble of 216 models. These models were then projected onto the statistically downscaled CCSM global climate model that is available from http://www.worldclim.org. All projections were converted to binary presence–absence maps using the equal sensitivity plus specificity threshold criterion. These models were further interrogated, using the SDMTools library version 1.1–13 (*VanDerWal et al., 2011*) in R3.0.1, by calculating the area and patch cohesion index within each catchment for each model combination within the ensemble for both current and past climate.

## RESULTS

### Genetic data characteristics

The final chloroplast dataset is comprised of 1,948 bp (trnQ-5′-rps16: 791 bp; atpI-atpH: 1,157 bp). The ITS and ncpGS datasets comprised 666 bp and 1,089 bp, respectively. Sequence characteristics for each dataset are summarised in Table 1. Haplotype/ribotype and nucleotide diversity were high in the ITS dataset, moderate in the chloroplast dataset, and low in the ncpGS dataset. Summaries of variable sites across the chloroplast and ncpGS haplotypes, as well as a subset of ITS samples, are shown in Appendix S3, respectively.

Two individuals situated in the Gamtoos had chloroplast haplotypes and ITS genotypes associated with the Sundays catchment lineage, however these were very close ($\leq$5 km) to the watershed suggesting incorrect watershed delineation or recent migration rather than incomplete lineage sorting. Thus, these samples were grouped with the Sundays basin samples for all analyses.

### Phylogeographic analyses

Under the evolutionarily discrete catchment (EDC) hypothesis, different catchments should contain genetically distinct lineages (Prediction 1). The catchment association was strong with chloroplast haplotypes, with all but one haplotype restricted to one of the primary or secondary catchments (Fig. 1B). A strong association was also evident between primary catchments and ITS clusters (Fig. 1D) with two exceptions (indicated with arrows, discussed further below). However, only rare ncpGS haplotypes were restricted to catchments, whereas two widespread haplotypes were shared either among both subregions of the Gouritz or among both Gamtoos and Sundays catchments (Fig. 1C). Two samples contained anomalous ncpGS haplotypes; these haplotypes are found in an intermediate position between the western and eastern haplotypes. They may represent inherited ancestral copy diversity or recent gene flow (subsequent intragenomic recombination) between samples from the western and eastern basins. However, given the slow rate of mutation in this region, evident by the low genetic diversity, and that one of these intermediate haplotypes is also shared with an outgroup sample from the northern distribution (BOL48535; Supplementary Files), it is likely that these haplotypes represent ancestral and unsorted ncpGS copies.
**Table 1  DNA summary statistics.**

| Genome | Catchment | $n$ | $h$ | $s$ | $\pi$ | $MT_R$ | $gsi_T$ |
|---|---|---|---|---|---|---|---|
| cpDNA | Overall | 78 | 20 | 28 | 0.0022 | 0.4510 *** | |
| | Groot[a] | 14 | 4 | 7 | 0.0015 | −0.0517 | 0.290 (0.081) * |
| | Olifants[a] | 26 | 8 | 11 | 0.0016 | 0.0638 | 0.508 (0.092) ** |
| | Gamtoos | 18 | 5 | 11 | 0.0008 | −0.0250 | **0.991** (0.067) ** |
| | Sundays | 18 | 7 | 8 | 0.0013 | 0.1077 | **0.683** (0.140) ** |
| | Gro + Oli[a] | 40 | 10 | 12 | 0.0017 | 0.0885 * | **0.947** (0.155) ** |
| | Oli + Gam | | | | | 0.3930 *** | 0.353 (0.097) ** |
| | Sun + Gam | 38 | 11 | 16 | 0.0017 | 0.3460 *** | **0.783** (0.119) ** |
| ITS | Overall | 75 | 68 | 54 | 0.0258 | 0.6234 *** | |
| | Groot[a] | 14 | 14 | 20 | 0.0128 | 0.1889 | 0.289 (0.025) ** |
| | Olifants[a] | 24 | 22 | 28 | 0.0133 | 0.1078 | 0.474 (0.039) ** |
| | Gamtoos | 15 | 13 | 23 | 0.0127 | 0.3433 * | **0.996** (0.018) ** |
| | Sundays | 22 | 21 | 21 | 0.008 | 0.1619 * | **0.994** (0.034) ** |
| | Gro + Oli[a] | 38 | 34 | 29 | 0.0133 | 0.0829 * | **0.945** (0.007) ** |
| | Oli + Gam | | | | | 0.6448*** | 0.456 (0.083) ** |
| | Sun + Gam | 37 | 34 | 35 | 0.0174 | 0.5622 *** | **0.990** (0.082) ** |
| ncpGS | Overall | 69 | 13 | 11 | 0.0025 | 0.5717 *** | |
| | Groot[a] | 13 | 3 | 4 | 0.0006 | −0.1778 | 0.251 (0.098) ** |
| | Olifants[a] | 22 | 5 | 3 | 0.001 | −0.1435 * | 0.393 (0.132) ** |
| | Gamtoos | 14 | 5 | 5 | 0.0016 | 0.2020 | 0.187 (0.060) ** |
| | Sundays | 20 | 2 | 1 | 0.0001 | −0.1236 | 0.546 (0.174) ** |
| | Gro + Oli[a] | 35 | 7 | 7 | 0.0009 | 0.0372 | **0.842** (0.284) ** |
| | Oli + Gam | | | | | 0.5454 *** | 0.000 (0.000) ** |
| | Sun + Gam | 34 | 6 | 6 | 0.0008 | 0.0875 | **0.848** (0.267) ** |

**Notes.**

$n$, number of samples; $h$, number of haplotypes; $s$, number of segregating sites; $\pi$, nucleotide diversity; $MT_R$, Mantel tests $r$ value (* $p < 0.05$; ** $p < 0.01$; *** $p < 0.001$); $gsi_T$, mean and standard deviation (in brackets) of the ensemble genealogical sorting index, and values above a threshold of 0.6 are in bold.

[a]The Groot and Olifants are sub-catchments within the primary Gouritz catchment that is referred to in the text.

Samples AJP0537 and AJP0810 (indicated with arrows in Figs. 1B–1D) lie within the Gamtoos catchment, but are close to sampled localities with a Sundays catchment genetic signature found on the watershed boundary between the catchments. These samples have Gamtoos chloroplast haplotypes, and for ncpGS, a rare Gamtoos haplotype, as well as a common and widespread haplotype. However, the first sample has an ITS signal that nests it within the Sundays ITS cluster 3, while the second displays a recombinant signal that lies between the Gamtoos cluster 2 and cluster 3 (Fig. 1D). The contrasting basin associations between these two samples suggest that they represent a contact zone between the Sundays and Gamtoos lineages, possibly caused by pollen flow between plants in this zone. As these samples are anomalous to the overall patterns of association, and are geographically restricted, they were removed for all subsequent analyses.

Isolation by distance using the Mantel Test (MT) was used to determine if there was genetic structuring within and between catchments (Prediction 2). In general, non-significant $MT_R$ values were observed within the primary or secondary catchments (Table 1)
across the different DNA regions; however, there were a few instances where significant but low $MT_R$ (ITS in Sundays; ncpGS in Olifants) and significant but high $MT_R$ values (in comparison to the overall value; ITS in Gamtoos) were observed. There were significant and high $MT_R$ values between the Olifants and Gamtoos basins across all DNA regions. Significant $MT_R$ values between the Groot and Olifants and the Gamtoos and Sundays were only observed in the cpDNA and ITS; however, the $MT_R$ values were low for the former pair of basins and high for the latter pair.

If watersheds have been evolutionarily long term barriers to gene flow then topographic-driven isolation should be evident, culminating in sorting, and ultimately monophyly, of lineages within catchments (Prediction 3). Although monophyletic lineages were detected with moderate to high support (60%–98%) in the chloroplast bootstrap phylogenies, only samples from the Gamtoos catchment displayed catchment-based monophyly (Fig. 1B). In contrast, the ITS produces high support for Gouritz, Gamtoos and Sundays lineages (Fig. 1D). No monophyletic lineages were found to be restricted to any catchments in the ncpGS data, but this is due to two widespread haplotypes and low diversification rates. Despite lack of supported monophyly in the cpDNA and ncpGS analyses, the *gsi* results do suggest a high level of lineage sorting (>0.600) in the western Gouritz basin for both datasets, and in the Gamtoos and Sundays for cpDNA (Table 1).

A BEAST analysis was used to date the diversification of chloroplast lineages in order to determine whether the Pleistocene climate cycles have also played a role in population isolation in catchments (Prediction 4). The timing of all of the thicket lineages fall firmly within the Pleistocene whether a 'fast' or 'slow' rate of chloroplast mutation is used (Fig. 2). Species distribution modelling suggests that *N. capensis* experienced major range contractions and fragmentation during the Last Glacial Maximum (Fig. 3, Prediction 5), finding refugia within each of the catchments and current distribution.

## DISCUSSION

In a broad sense, this study makes use of both genetic and geospatial data in order to explore $\beta$-clade formation driven by "orbitally-forced range dynamics" (ORD; *Jansson & Dynesius, 2002*) in a region that remained free of glaciation during the Pleistocene. More specifically, it tests the predictions of two ORD-related hypotheses regarding the effects of topography and Pleistocene climate fluctuations on the genetic diversity of *Nymania capensis* within the Albany Subtropical Thicket: the evolutionarily distinct catchment (EDC) hypothesis and the glacial multiple refugia (GMR) hypothesis. The former hypothesis was generated from a phytosociological study of thicket (*Vlok, Euston-Brown & Cowling, 2003*) where catchments were identified as discrete biogeographic units, whereas the latter was generated from biome-level distribution modelling (*Potts et al., 2013b*) and observations of physiological sensitivity to declining temperatures of thicket species (*Cowling, Proches & Vlok, 2005*; *Duker et al., 2015a*). From these hypotheses, five predictions were generated and tested using *Nymania capensis*. The results suggest that catchments have been *in situ* refugia (*Gavin et al., 2014*; *Tzedakis, 1993*) for *N. capensis* during glacial-interglacial cycles with glacial refugia largely located within the interglacial distribution.

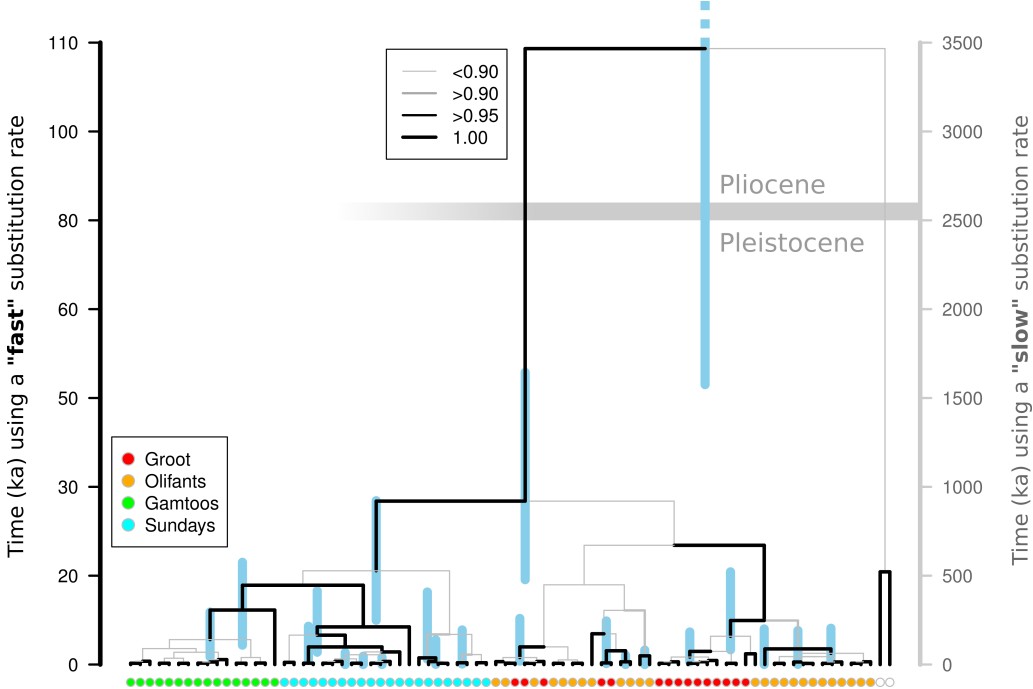

**Figure 2** **Molecular dating of *Nymania capensis* cpDNA sequences using majority rule Bayesian chronogram generated in Beast.** The posterior probabilities of branches are shown using a combination of branch width and colour. Nodes are centred on the median time to the most common recent ancestor (TMRCA) with blue shaded bars indicating the distribution of the 95% HPD for each estimate. The timing of divergence estimates are shown using a "slow" and "fast" substitution rate (see text for details). The horizontal grey bar indicates the Pliocene-Pleistocene boundary (∼2.6 Ma) under the "slow" substitution rate.

## Evolutionarily discrete catchments: watersheds and catchments as drivers of diversification

Investigating the role that catchment topography plays in reducing gene flow and driving population divergence has largely been restricted to either obligate freshwater animal species (e.g., redfins, *Swartz, Skelton & Bloomer, 2009*) or freshwater animal species capable of terrestrial movement (e.g., freshwater crayfish, crabs, salamanders, and spotted frogs; *Cook, Pringle & Hughes, 2008*; *Funk et al., 2005*; *Giordano, Ridenhour & Storfer, 2007*; *Ponniah & Hughes, 2006*). Only recently have catchments and watersheds been explored as drivers of diversification in terrestrial species that do not rely on the riparian system, and these have mostly focussed on invertebrates (e.g., springtails and cicadas; *Garrick et al., 2007*; *Garrick et al., 2004*; *Price, Barker & Villet, 2010*). The EDC hypothesis for the catchments in this study area is supported by two other studies, one on fish (*Swartz, Skelton & Bloomer, 2009*) and the other on cicadas (*Price, Barker & Villet, 2010*).

The phylogeographic patterns of *N. capensis* are largely consistent with the three predictions deduced from the EDC hypothesis. Firstly, the predominant pattern is one of genetically distinct lineages restricted to single catchments (Fig. 1), although a few lineages span neighbouring catchments. The number of genetically distinct lineages varies between the different markers; this is expected given their different rates of mutation and effective
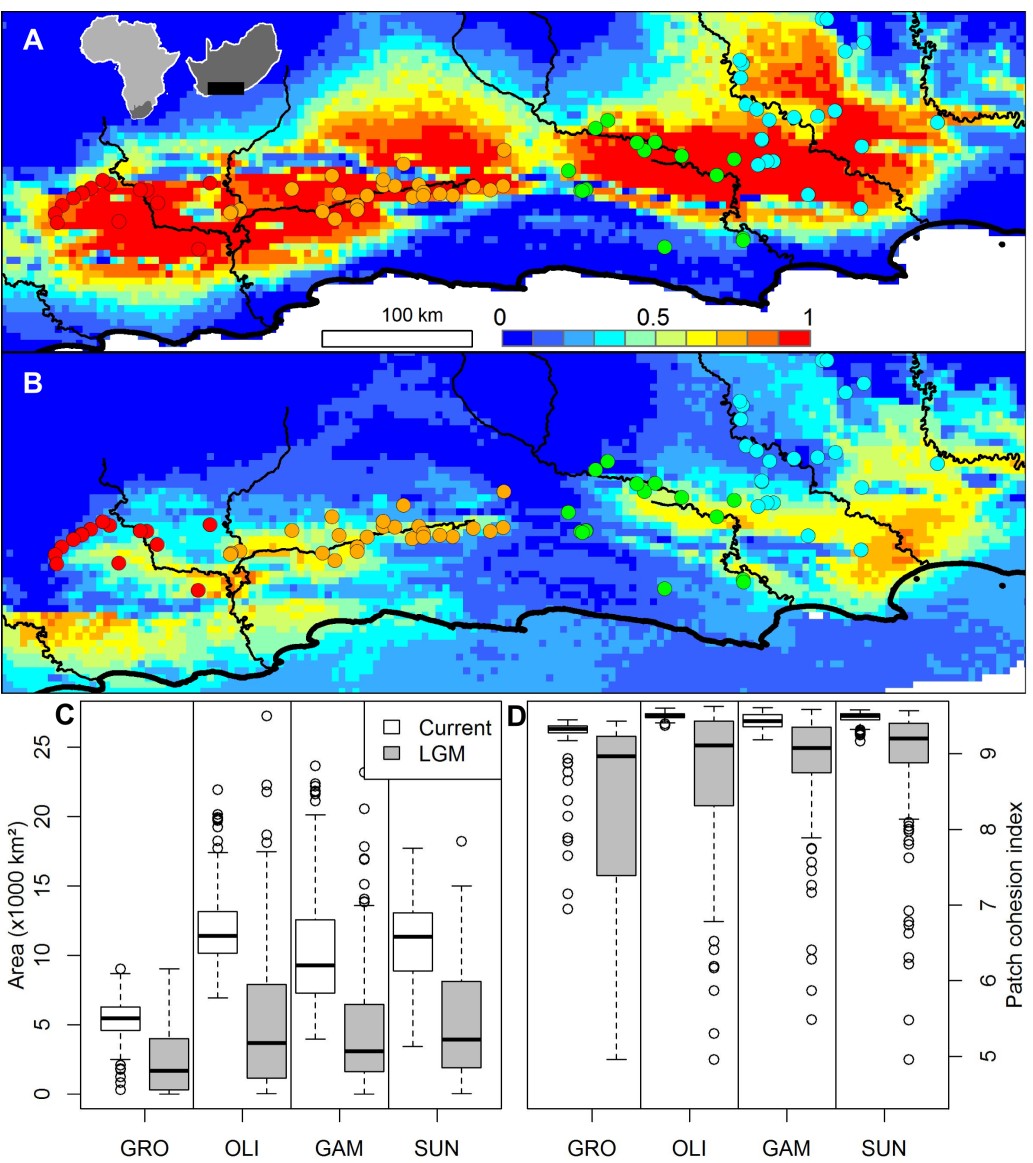

**Figure 3** The ensemble of 216 species distribution models of *Nymania capensis* projected onto (A) present and (B) Last Glacial Maximum (CCSM) simulated and downscaled climate, as well as the changes in climatically suitable (C) area, and (D) patch cohesion index, within each basin. In the distribution maps, red indicates greater certainty of presence, blue indicates greater certainty of absence, and green (~0.5) indicates areas of greatest model uncertainty.

population sizes. Secondly, there is limited genetic structuring within catchments as most analyses did not detect a within-basin signal of isolation by distance. In contrast, a large and significant isolation by distance effect is evident across all catchments (Table 1). This suggests that gene flow is hindered between basins, while this is not the case within basins. Lastly, although catchments do not contain genetically homogenised and fully isolated lineages, there is strong evidence of genealogical sorting within each catchment (Table 1).
The strong phylogeographic break between the western Gouritz and the eastern Gamtoos and Sundays catchments detected in this study corresponds to previously documented breaks observed within *Meleuphorbia* (Euphorbiaceae, *Ritz, Zimmermann & Hellwig, 2003*) and *Platypleura plumosa* (Hemiptera: Cicadidae, *Price, Barker & Villet, 2010*). The Rooiberg inselberg that separates the Groot and Olifants basin has been observed as a phylogeographic break in *Platypleura karooensis* (Hemiptera: Cicadidae, *Price, Barker & Villet, 2010*) and *Berkheya cuneata* (Asteraceae, *Potts et al., 2013a*). However, although only one chloroplast haplotype is shared between these subcatchments, which is suggestive of restricted seed flow, there is very weak support for isolation by distance or genealogical sorting across all DNA regions.

Thus, the results of this study provide additional support for the EDC hypothesis as the marked genetic structuring revealed in *N. capensis* is consistent with patterns predicted on the basis of primary catchment divisions. However, although the landscape has been stable since the Late Pliocene (*Cowling, Proches & Partridge, 2009*), dramatic cycling between glacial and inter-glacial climates has occurred during the Pleistocene. These shifts may have also affected the distribution, fragmentation and divergence of thicket species such as *Nymania capensis* (*Cowling, Proches & Vlok, 2005*; *Dynesius & Jansson, 2000*).

## Glacial refugia: Pleistocene climatic cycles as drivers of diversification

If the Pleistocene climate cycling between glacial and interglacial periods was responsible for isolating populations in this landscape—which has been stable since the late Pliocene ($\sim$2.6 Ma; *Cowling, Proches & Partridge, 2009*)—then it would be expected that lineage diversification would coincide or occur after the onset of these cycles. There are many potential problems with the molecular clock approach (*Graur & Martin, 2004*; *Ho, 2007*), and selecting an accurate rate of substitution for target taxa is of primary concern when fossil calibration is not possible. Using a wide range of published substitution rates for non-coding chloroplast DNA should circumvent the lack of an accurate rate for *Nymania capensis*. Using both slow and fast substitution rates, the divergence of all thicket lineages falls well within the Pleistocene (Fig. 2), suggesting that this species has experienced fragmentation and isolated diversification during this period. Under the fast substitution rate, many of the lineages diverge after the Last Glacial Maximum. This is, however, an unlikely scenario given that this is an exceptionally fast rate that has been derived from a genus of annual herbs (*Linum*) and the fast turnover in generations is likely to have greatly increased the substitution rate (*Kay, Whittall & Hodges, 2006*). *Nymania capensis* is a perennial plant that requires more than five years of ideal environmental conditions before flowers and seeds are produced (J Vlok, pers. comm., 2011). It is thus highly unlikely that its substitution rate would coincide with that of an annual herb, leading to the conclusion that lineage diversification fell within the Pleistocene. The slower substitution rate also falls within the generally accepted range of 1.0 to $3.0 \times 10^{-9}$ (*Wolfe, Li & Sharp, 1987*).

Climatic changes during the Pleistocene glacial cycles have induced distributional shifts in species, often resulting in fragmentation and divergence of populations (*Hewitt,*

*2004*; Fig. 3; *Jansson & Dynesius, 2002*). Identifying refugial areas during glacial periods through the Pleistocene has been a strong focus within many phylogeographic studies (*Avise, 2000*), with the majority of studies focussed on the previously glaciated northern hemisphere regions (*Abbott et al., 2000*; *Soltis et al., 1997*; *Taberlet et al., 1998*). Southern Africa did not experience glaciation during the Pleistocene climate cycles (*Partridge, 1997*) and determining refugia for plant species in areas that did not experience glaciation is a challenging process (e.g., *Byrne, 2007*). Nevertheless, in the Cape Floristic Region, which neighbours the thicket, climatic fluctuations during the Pleistocene have been suggested to be the main driver of fragmentation and shifts in faunal species distributions, culminating in allopatric diversification (e.g., rock agama, dwarf chameleons, and cicadas; *Price, Barker & Villet, 2007*; *Swart, Tolley & Matthee, 2009*; *Tolley et al., 2006*). The lower temperatures, both globally (*Zachos et al., 2001*) and regionally (*Holmgren et al., 2003*; *Talma & Vogel, 1992*), coupled with the frost sensitivity observed in many of the thicket's component species (*Duker et al., 2015a*; *Duker et al., 2015b*) is suggested to have driven thicket vegetation into fragmented refugia (*Cowling, Proches & Vlok, 2005*), as reflected by *Nymania capensis*.

Climatic changes will drive shifts in geographic distributions of species (*Jansson & Dynesius, 2002*) if they cannot adapt to new ecological conditions, at least over moderate periods of time. Evidence suggests that niche conservatism may be general and pervasive across most species over moderate periods of time, despite profound changes in climate and environmental conditions (*Martínez-Meyer & Peterson, 2006*; *Martínez-Meyer, Peterson & Hargrove, 2004*; *Peterson, Soberon & Sanchez-Cordero, 1999*). Here I assume that the niche of *N. capensis* has been largely conserved from the LGM to the present.

The species distribution modelling results of *N. capensis* suggest this species' range contracted and fragmented into and within the primary catchments (Fig. 3). This is consistent with the phylogeographic evidence that suggests isolation into at least three refugia which correspond to the delimitation of primary basins (Fig. 1). Thus, the retraction into catchments during glacial periods through the Pleistocene would have strengthened the effects of watershed barriers to gene flow. This retraction is consistent with the limited palaeodata (reviewed in *Cowling, Proches & Vlok, 2005*) and community distribution modelling of thicket subtypes (*Potts et al., 2013b*) that also suggests that thicket suffered significant range constrictions during the most recent Pleistocene glacial period. Also, the absence of vertebrates endemic to the thicket, which would be expected given the present-day area of the biome, is suggestive of historical reduction and fragmentation of the biome's distribution (*Hoare et al., 2006*).

The EDC hypothesis has been used as the cornerstone for conservation planning in the Albany Subtropical Thicket in order to conserve both the biodiversity patterns and evolutionary processes of this vegetation (*Rouget et al., 2006*). Specifically, a number of conservation corridors have been identified to create a mega-conservancy network; these corridors are predominantly focussed on conserving major environmental gradients primarily within catchments. These results demonstrate that the patterns observed in the chloroplast phylogeography (*Potts, Hedderson & Cowling, 2013*) are mirrored in the nuclear genome, suggesting that the catchments house evolutionarily significant units (sensu *Crandall et al., 2000*; *Moritz, 1994*).

In conclusion, the results from these phylogeographic analyses and ensemble niche modelling suggest that the genetic structuring of *N. capensis* has been determined by landscape topology coupled with the effects of fluctuating Pleistocene climates. Populations have been restricted to three primary catchments with no extensive present-day or historical gene flow during the previous glacial period. This is largely consistent with the predictions derived from a scenario where landscape topography and climatic fluctuations were responsible for structuring populations across the southern African lowlands. From their glacial refugia they (re-)colonised their modern range, a process that is apparently accompanied by secondary contact and introgression (seen in the 'anomalous' Gamtoos samples). The findings validate and further highlight the importance of including catchments in conservation planning as discrete and evolutionarily significant units important for maintaining genetic resources.

## ACKNOWLEDGEMENTS

The author gratefully acknowledges Richard M. Cowling, Terry A. Hedderson, Guido W. Grimm, and three anonymous reviewers for comments that greatly improved this manuscript. In addition, Jan Vlok is thanked for sharing his knowledge regarding his extensive natural history observations of this species and the thicket vegetation.

### Funding

This research was supported by Darwin Initiative, National Research Foundation and Claude Leon Foundation. The funders had no role in study design, data collection and analysis, decision to publish, or preparation of the manuscript.

### Grant Disclosures

The following grant information was disclosed by the author:
Darwin Initiative, National Research Foundation, Claude Leon Foundation.

### Competing Interests

The authors declare there are no competing interests.

### Author Contributions

- Alastair J. Potts conceived and designed the experiments, performed the experiments, analyzed the data, contributed reagents/materials/analysis tools, wrote the paper, prepared figures and/or tables, reviewed drafts of the paper.

### Field Study Permissions

The following information was supplied relating to field study approvals (i.e., approving body and any reference numbers):
Sampling permits were obtained from the Eastern Cape Parks and Tourism Agency and CapeNature (0028-AAA008-00015).

## DNA Deposition

The following information was supplied regarding the deposition of DNA sequences:

These have been uploaded as a supplemental file (alignments) and have been lodged with GenBank.

Genbank rps16-trnQ intergenic spacer: KF180321–KF180398.

Genbank atpH-atpI intergenic spacer: KF180399–KF180476

Genbank internal transcribe spacer 1, 5.8S ribosomal RNA gene, internal transcribed spacer 2: KF443002–KF443033, KY095713–KY095757.

Genbank ncpGS gene: KY095758–KY095827.

## Data Availability

The raw data, including DNA sequence alignments, has been supplied as a Supplementary File.

## Supplemental Information

Supplemental information for this article can be found online at http://dx.doi.org/10.7717/peerj.2965#supplemental-information.

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
