# Peer review of "Catchments catch all in South African coastal lowlands: topography and palaeoclimate restricted gene flow in Nymania capensis (Meliaceae)—a multilocus phylogeographic and distribution modelling approach"

_PeerJ, doi:10.7717/peerj.2965_

## Round 0.1 · original submission · Major Revisions

· Academic Editor

Major Revisions

Dear author

As you can see one of the reviewers requests some changes. If you are able to submit a thorough revision we will send your ms to the original reviewers again.

Greetings
Michael Wink
Academic editor

Reviewer 1 ·

Basic reporting

The manuscript is well-written and interesting. It presents data which support the importance of catchments in conservation of the Albany thickets of South Africa. All figures and supporting documents are of a suitable quality.

Experimental design

Experimental design was described in detail and references were cited to support most aspects of the analyses.

Validity of the findings

The hypotheses tested are clearly stated and the methods address these. The results lend some support towards these hypotheses, with some minor exceptions. The main aspects tested are supported by the data presented.

Additional comments

The author is congratulated on a well constructed and interesting manuscript.

·

Basic reporting

The paper by Potts analysed the genetic variation, the divergence times and the distribution model of Nymania capensis.

I regard the ms. as interesting and the used methods as mostly adequate. The results show a geographic pattern, likely triggered by Pleistocene climatic cycles. I think that the ms. is well done, I mainly suggest to add another approach to the dating method. Therefore, I officially recommend major revisions to this ms.

Beside the methodological point, I have some questions about the frame of the study.

The EDC hypothesis is very interesting. I think, biological systems, in special on the meta-species level mostly don’t act in ‘clear boundaries’ and are quite dynamic.

Despite several organisms have been analysed, is Nymania really representative for such a thicket pattern? In other words, is it justified to infer from this example while other taxa may show completely different ones? Biomes are usually – as far as I know – composed of organisms with different evolutionary histories.

What are the mechanisms that separate the populations of the different catchment areas apart? Are the taxa restricted to their catchment (e.g., physiologically), is there limited exchange, pollination and dispersal ability, competition/ establishment factors, reproductive barriers (less likely)?
What other factors could be relevant?

The differentiation btw. East and West pop. is nicely reflected in the LGM climatic reconstruction. However, within the blue/green and red/yellow areas there is less. What about repeated retreat to relict areas?
What could have happened before the Late Pleistocene (in the basal crown branching)?
Could it be a pure east-west gradient or a central origin with extension to the periphery, with following partial geographic differentiation?

How fast do watersheds change? In Central Europe water systems may be geographically +/- stable for the Pleistocene, but definitely changed drastically since the Upper Miocene. Is there geological evidence that the catchment areas were indeed relatively stable throughout the Pleistocene in this area?

Beast: As a first approach it is feasible to use substitution rates. I would strongly recommend the author to add another step. You should use Magallon 2015 (New Phytol.) family ages 66.38- 73.69 my 95HPD for Meliaceae as uniform & Yule priors in a reduced Meliaceae taxon set incl. Nymania capensis and its closest relative. In a 2nd step, this age range can be applied to an infraspecific set of N.c. (incl. sister) under a coalescent model. This could provide a narrower dating frame, and this approach is better supported by data attributed to Meliaceae.



Minor points:
- l. 14. orbitally-forced range dynamics, I would better refer to climatic cycles.
- l. 30 evidence for seed and pollen flow?
- l. 146 evidence for point iii?

Mike Thiv

Experimental design

See Basic reporting.

Validity of the findings

See Basic reporting.

Additional comments

See Basic reporting.

---

## Round 0.2 · Major Revisions

· Academic Editor

Major Revisions

Dear author

This is your last chance to implement the request of the reviewer. If you are not willing to do so, I have to reject your ms.

Hoping for your understanding

Michael Wink
Academic editor

·

Basic reporting

Dear Dr Potts,

I do not appreciate the tone of your ‘rebuttal’.
I asked you several questions about the EDC hypothesis. My intension was not to offend you and I did not expect you to test all these factors, but the aim was to see how it is linked to other systems (which I think is very interesting). If a hypothesis cannot be (easily) linked to other biological systems it is not of much biological relevance.
For example, the question whether Nymania capensis is really representative to infer evolutionary scenarios for the higher biome or vegetation type, respectively, was not answered using arguments.
Dating method: There is no right or wrong dating method. All approaches have their advantages and disadvantages. I am aware that you lose accuracy by using a 2 step approach. Applying substitution rates from ACROSS ALL angiosperms to a Meliaceae taxon is ‘really broad’ and varies among the factor 50 (Kay et al 2006). Afterwards arguing that some are too fast may be likely, but is not proven. By adding a second approach you could have discussed both outcomes and even include some critiques as mentioned by Sauquet et al. Such approach has been used in several papers, also after this paper. In total, comparing 2 approaches is more accurate and interesting than only applying substitution rates. Moreover, it is one of the main criteria you try to test your EDC hypothesis with. By the end it is most likely, almost trivial, that most of the infraspecific genetic variation in plants evolved in the Pleistocene as shown for many worldwide distributed, different taxa.
My advice is that the ms. would indeed benefit from a second dating approach. Therefore I keep major revisions to this ms. If the author is not willing to implement a Meliaceae calibrated dating, I ask the editor to handle this ms. over to another reviewer.
NB. No need to mention me in the acknowledgements.
Best wishes
Mike Thiv

Experimental design

see Basic Reporting

Validity of the findings

see Basic Reporting

Additional comments

see Basic Reporting

---

## Round 0.3 · Minor Revisions

· Academic Editor

Minor Revisions

Dear authors

Your ms has been reviewed again by another reviewer who only suggests a minor revision. But please take the criticism from the first original reviewer into account.

Regards
Michael Wink

Reviewer 3 ·

Basic reporting

no comment

Experimental design

no comment

Validity of the findings

no comment

Additional comments

I was explicitly asked to comment on the dating issue:

I agree with Mike that no other than a Pleistocene differentiation scenario would have been expected in an intraspecific study. Due to a lack of reliable calibration points the author solely worked with substitution rates. But he applies both a fast and a slow rate that both result in median ingroup ages in the Pleistocene (even including HPD). Since this is only one aspect of this single-author study, I agree with the author that there is no need to incorporate further - questionable - dating approaches.

I am concerned that in Fig. 2 fully supported nodes (1.0) are not marked. Since in Bayesian phylogenetic approaches the threshold for node support lies at 0.95, the four levels of line thickness are very strange to me.

In the final version, Fig. 2 should be upright.

---

## Round 0.4 · accepted · Accept

· Academic Editor

Accept

Dear Alastair

Thanks for the revisions. Your ms can be accepted now

Regards,

Michael Wink
Academic editor